# Oral administration of pyrophosphate inhibits connective tissue calcification

Dóra Dedinszki[1,†], Flóra Szeri[1,†], Eszter Kozák[1,2], Viola Pomozi[1,3], Natália Tőkési[1], Tamás Róbert Mezei[4,5], Kinga Merczel[1], Emmanuel Letavernier[6,7,8], Ellie Tang[6,7], Olivier Le Saux[3], Tamás Arányi[1,9], Koen van de Wetering[10,‡] & András Váradi[1,*] 

## Abstract

Various disorders including pseudoxanthoma elasticum (PXE) and generalized arterial calcification of infancy (GACI), which are caused by inactivating mutations in *ABCC6* and *ENPP1*, respectively, present with extensive tissue calcification due to reduced plasma pyrophosphate (PPi). However, it has always been assumed that the bioavailability of orally administered PPi is negligible. Here, we demonstrate increased PPi concentration in the circulation of humans after oral PPi administration. Furthermore, in mouse models of PXE and GACI, oral PPi provided via drinking water attenuated their ectopic calcification phenotype. Noticeably, provision of drinking water with 0.3 mM PPi to mice heterozygous for inactivating mutations in *Enpp1* during pregnancy robustly inhibited ectopic calcification in their *Enpp1*$^{-/-}$ offspring. Our work shows that orally administered PPi is readily absorbed in humans and mice and inhibits connective tissue calcification in mouse models of PXE and GACI. PPi, which is recognized as safe by the FDA, therefore not only has great potential as an effective and extremely low-cost treatment for these currently intractable genetic disorders, but also in other conditions involving connective tissue calcification.

**Keywords** generalized arterial calcification of infancy; oral pyrophosphate treatment; pseudoxanthoma elasticum; soft tissue calcification
**Subject Categories** Musculoskeletal System; Pharmacology & Drug Discovery

## Introduction

Physiological mineralization is essential for the normal development of vertebrates and restricted to specific sites of the body. In mammals, biominerals predominantly consist of calcium and phosphate, which form hydroxyapatite. Vertebrates have evolved mechanisms permitting crystallization of calcium and phosphate only at specific sites.

Pyrophosphate (PPi) is a central factor in the prevention of the precipitation of calcium and phosphate in soft peripheral tissues (for a recent review see: Orriss *et al*, 2016). The liver is the most important source of circulatory PPi, via a pathway depending on ABCC6-mediated ATP release (Jansen *et al*, 2013, 2014), though the exact molecular mechanism of ATP release and the actual substrate of ABCC6 is not known. Within the liver vasculature, released ATP is rapidly converted into AMP and PPi by the ectoenzyme ectonucleotide pyrophosphatase phosphodiesterase 1 (ENPP1; Jansen *et al*, 2014). Inactivating mutations in the genes encoding enzymes involved in PPi homeostasis result in rare hereditary calcification disorders which include pseudoxanthoma elasticum (PXE, OMIM 264800), generalized arterial calcification of infancy (GACI, OMIM 208000), arterial calcification due to CD73 deficiency (ACDC, OMIM 211800), and Hutchinson–Gilford Progeria Syndrome (HGPS, OMIM 176670). Absence of functional ABCC6 results in PXE, a late onset ectopic calcification disorder, with lesions found in the skin, eyes, and cardiovascular system (Bergen *et al*, 2000; Le Saux *et al*, 2000; Ringpfeil *et al*, 2000). Biallelic mutations in *ENPP1* cause GACI, a condition that can become life threatening shortly after birth due to massive calcification of the large- and medium-sized arteries (Rutsch *et al*, 2003). While in PXE plasma PPi concentration

1  Institute of Enzymology, RCNS, Hungarian Academy of Sciences, Budapest, Hungary
2  Department of Immunology, ELTE, Budapest, Hungary
3  Department of Cell and Molecular Biology, John A. Burns School of Medicine, University of Hawaii, Honolulu, HI, USA
4  Department of Mathematics and its Applications, Central European University, Budapest, Hungary
5  Alfréd Rényi Institute of Mathematics, Hungarian Academy of Sciences, Budapest, Hungary
6  Sorbonne Universités, UPMC Univ Paris 06, UMR S 1155, Paris, France
7  INSERM, UMR S 1155, Paris, France
8  Physiology Unit, AP-HP, Hôpital Tenon, Paris, France
9  MITOVASC, CNRS UMR 6015, Inserm U1083, University of Angers, Angers, France
10 Division of Molecular Oncology, Netherlands Cancer Institute, Amsterdam, The Netherlands
   *Corresponding author. Tel: +36 203 569135; E-mail: varadi.andras@ttk.mta.hu
   †These authors contributed equally to this work
   ‡Present address: Department of Dermatology and Cutaneous Biology, PXE International Center of Excellence in Research and Clinical Care, Sydney Kimmel Medical College at Thomas Jefferson University, Philadelphia, PA, USA

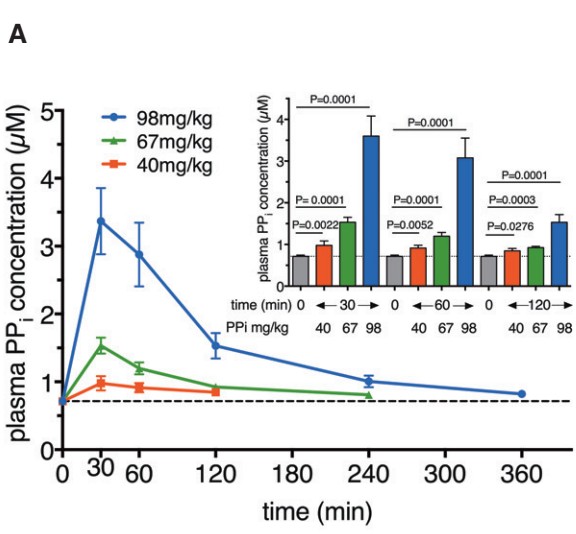

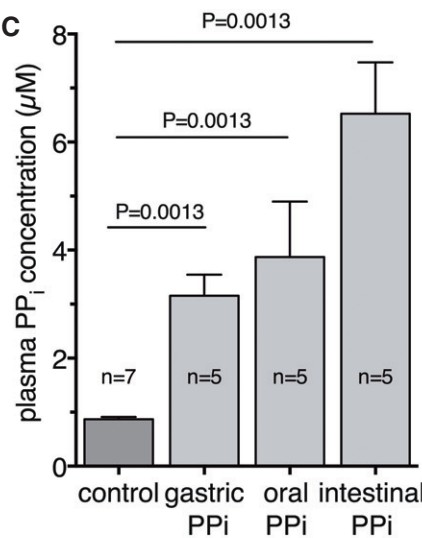

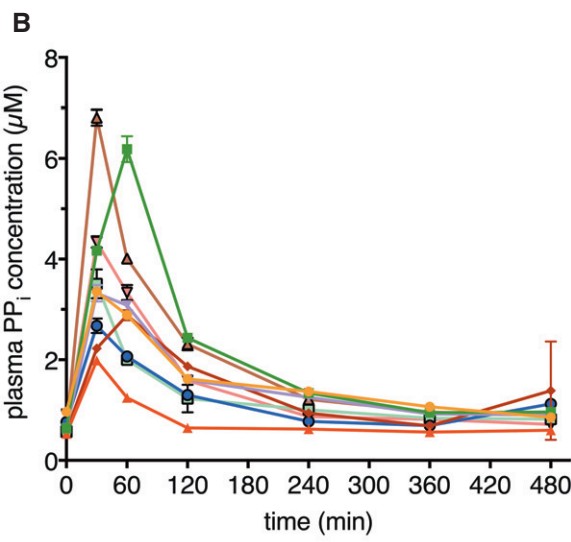

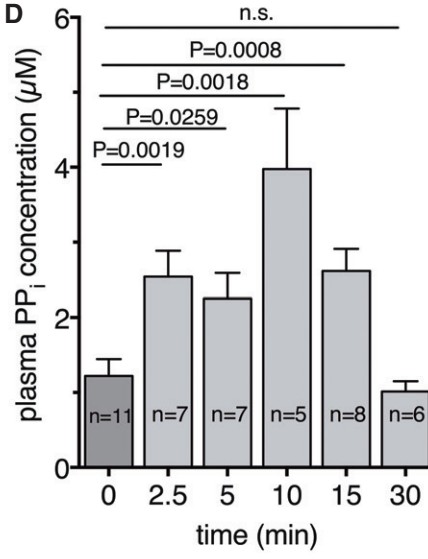

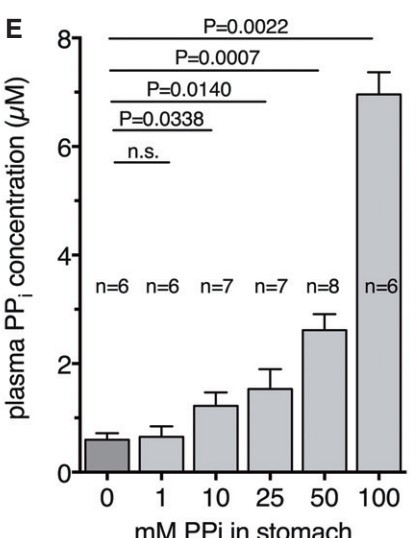

**Figure 1.**

◄ **Figure 1.  Uptake of PPi from drinking water.**

A  Oral uptake of tetrasodium pyrophosphate in humans. Volunteers ingested tetrasodium pyrophosphate solutions of 43, 72, 110 mM, pH 8.0, resulting in a dose of 40 mg/kg ($n$ = 10) or 67 mg/kg ($n$ = 10) or 98 mg/kg ($n$ = 9), respectively. Plasma PPi levels were determined before (0 min), and 30, 60, 120, 240, and 480 min after ingestion. The insert shows the differences between the basal plasma PPi level (0 min) and that 30, 60, and 120 min after ingestion.

B  Oral uptake of 98 mg/kg tetrasodium pyrophosphate ($n$ = 9) in human indicating individual differences.

C  Uptake from the oral cavity ($n$ = 5), from the stomach ($n$ = 5), and from the intestine ($n$ = 5) of C57/Bl6 mice after ligating the esophagus and applied oral delivery of 100 μl 50 mM PPi; or the pylorus followed by stomach delivery of 200 μl 50 mM PPi and then ligation of the esophagus; or injecting 200 μl 50 mM PPi into the intestine after ligation of the pylorus. In each experiment, including control ($n$ = 7), blood was collected after 15 min and PPi concentrations were determined.

D  Time-course of PPi uptake from the stomach of C57/Bl6 mice upon gavage delivery of 200 μl of 50 mM PPi, $n$ = 5–11.

E  Dose-dependent PPi uptake from the stomach of C57/Bl6 mice upon gavage delivery of 200 μl of PPi of concentration 0 ($n$ = 6), 1 ($n$ = 6), 10 ($n$ = 7), 25 ($n$ = 7), 50 ($n$ = 8), and 100 ($n$ = 6) mM. Blood was collected for PPi assay after 15 min of delivery.

Data information: Data were analyzed by two-tailed Mann–Whitney nonparametric test. Results are expressed as mean ± SEM.

is reduced to 40% of normal (Jansen *et al*, 2014), GACI patients have virtually no PPi in their blood, which explains the severity of the disease (Rutsch *et al*, 2003). Other gene products are also involved in soft tissue calcification affecting PPi homeostasis, such as ANK, which mediates the intracellular to extracellular channeling of PPi (Ho *et al*, 2000), though it does not play a role in maintaining plasma PPi.

Because reduced PPi concentrations in the circulation underlie the ectopic calcification disorders PXE and GACI, a logical treatment for these disorders would be PPi supplementation. Due to the necessity to treat patients lifelong, oral administration would be preferred for such a treatment. Phosphatases are abundantly present in the gut (Ferguson *et al*, 1968); therefore, it has been always claimed that orally administered PPi cannot reach the circulation and therefore is not effective in inhibiting ectopic calcification (Orriss *et al*, 2016). We have tested this assumption in healthy human individuals and in mouse models reflecting two human hereditary calcification disorders, PXE and GACI.

## Results

### Uptake of PPi in humans and mice

First, we tested whether orally consumed PPi is absorbed in humans. Healthy human volunteers (fasting) ingested a solution of tetrasodium pyrophosphate (Na$_4$PPi), resulting in a dose of 40, 67, or 98 mg/kg of body weight (43, 72, 110 mM, pH 8.0, respectively). The ingested amounts of Na$_4$PPi correspond to 13–33% of the maximal tolerable daily intake published by the World Health Organization, WHO (http://www.inchem.org/documents/jecfa/jeceval/jec_2259.htm). This resulted in substantially elevated plasma PPi levels (Fig 1A; with individual differences as shown on Fig 1B) when 98 and 67 mg/kg Na$_4$PPi was taken, while the plasma PPi level was increased only moderately in individuals taking 40 mg/kg (Fig 1A). The time needed to get the plasma PPi back to the baseline level was 240 min at dose 67 mg/kg and 360 min at dose 98 mg/kg (Fig 1A). These data indicate a dose- and time-dependent elevation of plasma PPi concentration. From the data presented in Fig 1A, we calculated the following pharmacokinetic parameters: $t_{max}$ = 36.7 ± 13.2 min, $C_{max}$ = 3.9 ± 1.6 μM, $t_{1/2}$ = 44.7 ± 16.7 min (single exponential decay) when 98 mg Na$_4$PPi per kg body weight was given.

Next, we investigated whether supplementation via the drinking water increased plasma PPi levels in mice. In preliminary experiments, we found that in half of the animals, plasma PPi did not increase. This is most probably due to "non-synchronized" drinking by the animals. We therefore directly delivered PPi into the oral cavity, stomach, or small intestine (Fig 1C) and placed ligatures downstream and/or upstream of the site of administration to prevent any transfer of PPi. Unexpectedly, PPi was remarkably well absorbed from all sites tested and we followed the uptake of PPi (50 mM, 200 μl) delivered directly to the stomach over time. PPi was rapidly absorbed from the stomach (Fig 1D) and, as expected, its plasma concentrations depended on the dose given (Fig 1E).

### Oral PPi inhibits ectopic calcification in Abcc6$^{-/-}$ mice

*Abcc6*$^{-/-}$ mice recapitulate human PXE, with calcifying lesions found in skin, eyes, and blood vessels (Gorgels *et al*, 2005; Klement *et al*, 2005). A drawback of the *Abcc6*$^{-/-}$ mouse model is the relatively late onset of the first symptoms, making it less convenient for rapid screening of new treatments. However, cryo-injury applied to the heart results in calcified lesions within 3–6 days (Doehring *et al*, 2006), a phenomenon fully dependent on the absence of *Abcc6* (Brampton *et al*, 2014). The lesions showed hydroxyapatite crystal nature as determined by transmission electron microscopy (Aherrahrou, 2003). Next, we determined whether orally administered PPi was able attenuating induced cardiac calcification in PXE mice. PPi provided via the drinking water potently inhibited calcification (PPi was provided in the drinking water a day before the cryo-injury and continued for 4 days) as shown by the reduced calcified area (Fig 2A, calcium deposits stained by Alizarin Red). The extent of inhibition was dose-dependent, with maximal inhibition seen at a PPi concentration of 10 mM, though the cardiac lesions in *Abcc6*$^{-/-}$ mice receiving drinking water with 1mM PPi already contained more than twofold less calcium (Fig 2B).

A hallmark phenotype seen in *Abcc6*$^{-/-}$ mice is the spontaneous gradual calcification of the connective tissue surrounding the vibrissae (Klement *et al*, 2005). We found that providing *Abcc6*$^{-/-}$ mice PPi-containing drinking water (10 mM) after weaning received for 19 weeks greatly reduced the extent of calcification found in the tissue surrounding the vibrissae. These data show that orally administered PPi not only inhibits the calcification seen after the application of cryo-injury, but also potently inhibits the spontaneous calcification seen in *Abcc6*$^{-/-}$ mice (Fig 2C–E).

### Oral PPi attenuates calcification in Enpp1$^{-/-}$ (ttw) mice

Tiptoe walking (*ttw*) mice have an inactivating mutation in *Enpp1* (Okawa *et al*, 1998). Due to the complete absence of Enpp1, these

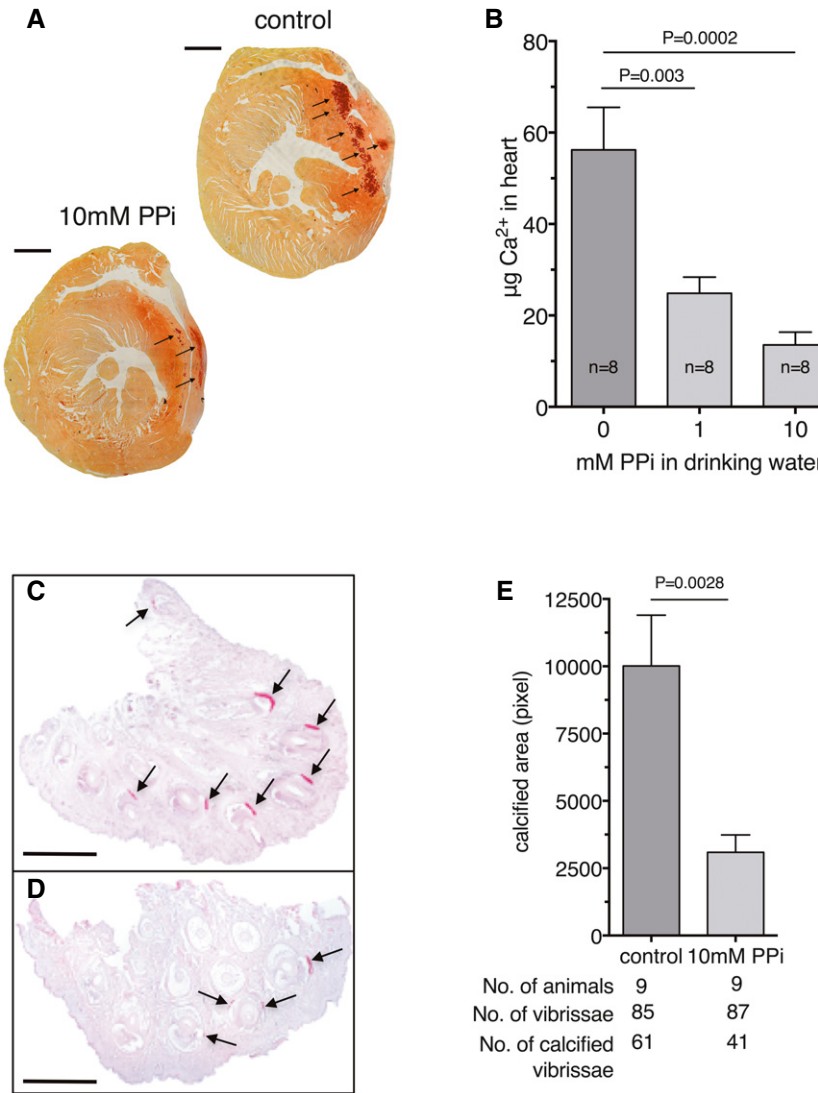

**Figure 2. Oral PPi attenuates induced cardiac and spontaneous calcification of vibrissae in Abcc6$^{-/-}$ mice.**

A Calcification of the heart of *Abcc6*$^{-/-}$ mice after 4 days of cryo-injury (drinking water, upper image) or (drinking water with 10 mM PPi, lower image). Ca-precipitations are indicated by arrows, and calcium deposits were visualized by Alizarin Red staining. Scale bar = 1 mm.

B PPi was provided in 0 (*n* = 8), 1 (*n* = 8) or in 10 (*n* = 8) mM concentrations via the drinking water to *Abcc6*$^{-/-}$ mice starting a day before cryo-injury for a total of 4 days. The calcium content of heart tissue was determined by complexometry.

C, D Show typical Alizarin Red-stained sections of an animal of the control group and that of the 10 mM PPi group, respectively. *Abcc6*$^{-/-}$ mice were kept on 10 mM PPi (in drinking water) starting at an age of 3 weeks (after weaning) until they were 22 weeks old. The control group was drinking tap water. Tissue blocks with the vibrissae were removed, paraffin-embedded, sectioned, and stained with Alizarin Red for calcium deposits, scale bar = 1 mm.

E The extent of calcification, control (*n* = 9) and 10 mM PPi (*n* = 9), was quantified by morphometry as described in the Materials and Methods.

Data information: Data were analyzed by two-tailed Mann–Whitney nonparametric test. Results are expressed as mean ± SEM.

mice have extremely low plasma PPi levels and like GACI patients, which develop extensive calcifications in blood vessels and joints shortly after birth (Nitschke & Rutsch, 2012). Just like *Abcc6*$^{-/-}$ mice, the *Enpp1*$^{-/-}$ mice develop extensive calcification of the dermal sheet surrounding the vibrissae. In these animals, this phenotype shows up much earlier than in the PXE mice, however.

Orally administered PPi during pregnancy and breastfeeding followed by oral PPi treatment of the pups, resulted in reduced calcification of the dermal sheet surrounding the vibrissae in the *Enpp1*$^{-/-}$ mice (see Fig 3A–C). We found that treatment of heterozygous Enpp1$^{+/-}$ mothers during their pregnancy with PPi was critical to inhibit the ectopic calcification seen in their Enpp1$^{-/-}$ offspring: Treating Enpp1$^{-/-}$ pups only after weaning did not attenuate ectopic calcification (Fig 3A). We detected a robust effect in the extent of calcification inhibition in the hind limb arteries and in the renal arteries. When PPi in as low as 0.3 mM concentration was provided during pregnancy, calcification was reduced to 12% of the levels found in the control group (hind limb arteries) and to 25% (renal arteries), that is, resulted in 75–88% inhibition (Fig 3D–I).

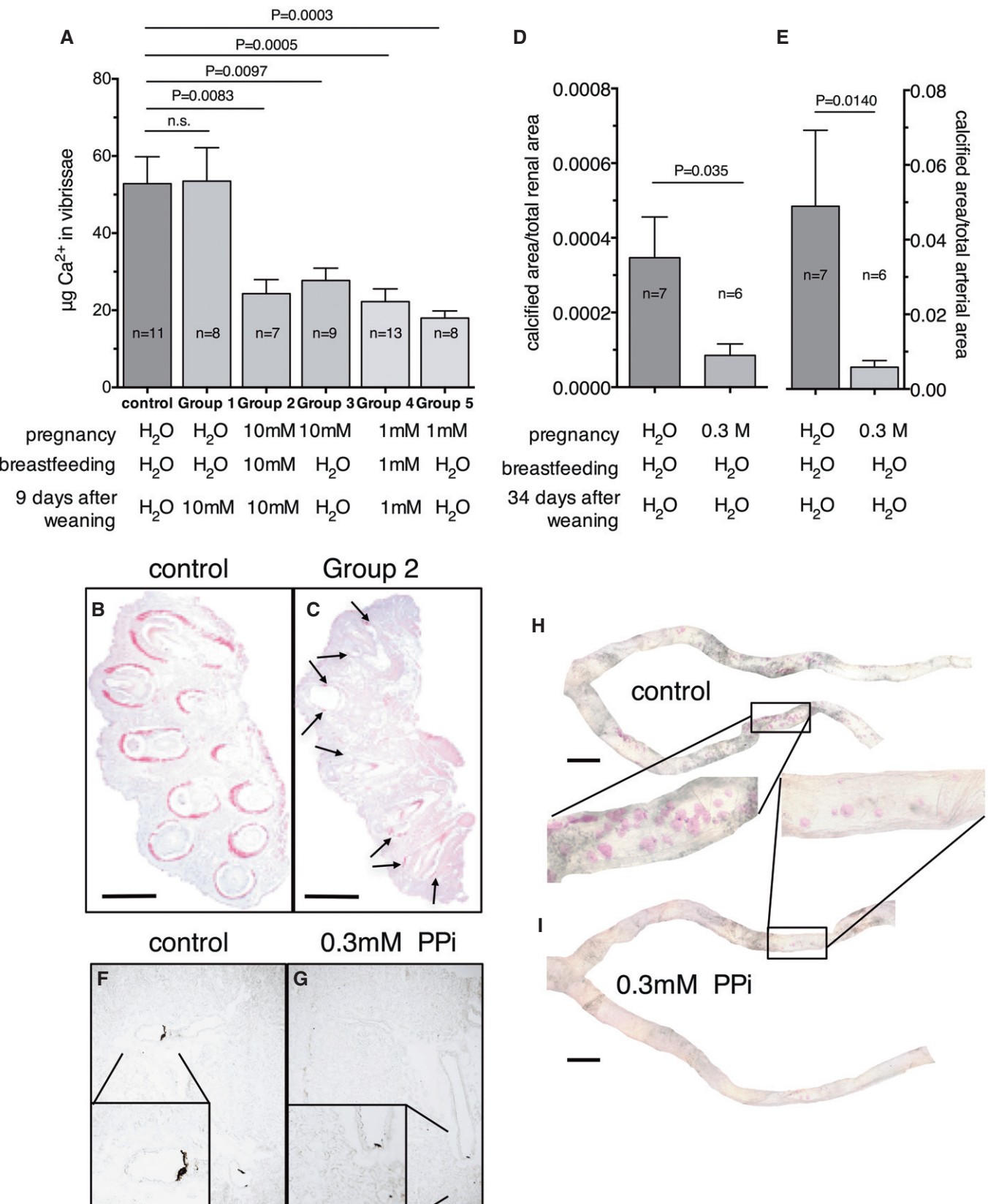

Figure 3.

◄

**Figure 3.  Prenatal PPi treatment of the Enpp1$^{-/-}$ mice attenuates calcification of the tissue surrounding the vibrissae and the arteries of the hind limbs and kidneys.**

A    Calcium content of the tissue blocks of the vibrissae. The heterozygous mothers and their offspring were kept on tap water until the pups were 30 days old ("control", n = 11); Group 1: as the control group, but for 9 days on 10 mM PPi solution after weaning at day 21 (n = 8); Group 2: the mothers and the pups were kept on 10 mM PPi during pregnancy, breastfeeding and for 9 days after weaning (n = 7). Group 3: The mothers were kept on 10 mM PPi during pregnancy (n = 9). Group 4: The mothers and the pups were kept on 1 mM PPi during pregnancy, breastfeeding, and for 9 days after weaning (n = 13). Group 5: The mothers were kept on 1 mM PPi during pregnancy (n = 8).

B, C    Typical Alizarin Red-stained sections of tissue blocks with the vibrissae of animals of different groups. Scale bar: 1 mm

D    Calcification of renal arteries. The *Enpp1$^{+/-}$* mothers were kept on tap water (n = 7) or on 0.3 mM PPi (n = 6) during pregnancy. Offspring was kept on tap water for 55 days.

E    Calcium content of the hind limb arteries. The heterozygous mothers were kept on tap water (n = 7) or 0.3 mM PPi (n = 6) during pregnancy. Offspring was kept on tap water for 55 days.

F, G    Typical kidney sections of a 55-day-old animal of the control group and of a 55-day-old animal from the group of 0.3 mM treatment only during pregnancy. Sections were stained by the Yasue procedure. Scale bar: 200 μm.

H, I    Typical Alizarin Red-stained hind limb arteries of a 55-day-old animal of the control group and those of a 55-day-old animal from the group of 0.3 mM treatment only during pregnancy. Scale bar: 1 mm.

Data information: Data were analyzed by two-tailed Mann–Whitney nonparametric test. Results are expressed as mean ± SEM.

## Discussion

Contrary to the general assumption, we found that PPi has bioavailability in humans and mice when administered orally (Fig 1). The observed transient elevation found in healthy volunteers (Fig 1A) indicates that in both GACI and in PXE patients, PPi levels may be transiently raised to the physiological level when 67 or 98 mg Na$_4$PPi/kg of body weight is taken. Importantly, it has been shown that inhibition of calcification in uremic rats and in PXE mice can be achieved by daily intraperitoneal injections of PPi triggering transient increase in plasma PPi levels (O'Neill *et al*, 2011; Pomozi *et al*, 2017). We determined a $t_{1/2}$ = 44.7 min what is rather similar to that published in rat (34.1 min; O'Neill *et al*, 2011).

We also demonstrated significant attenuation of calcification in two different well-characterized mouse models of ectopic calcification disorders, PXE and GACI when the animals were treated with pyrophosphate orally. Furthermore, we demonstrated PPi uptake from the oral cavity and stomach of mice allowing a portion of PPi to escape the rapid hydrolytic decomposition presumably occurring in the intestinal tract.

The unexpected observation that 0.3 mM PPi administration was highly effective if applied exclusively during pregnancy (see Fig 3) is likely due to microcrystal formation in the control-treated group before birth. These microcrystals might be mostly absent in *Enpp1$^{-/-}$* pups from heterozygous mothers receiving PPi during pregnancy and would therefore not be available to accelerate the calcification process after birth. GACI is often already diagnosed prenatally in the third trimester or at birth (Kalal *et al*, 2012) when the calcification is already present. The results in the *Enpp1$^{-/-}$* mice suggest that under these conditions oral PPi may not be effective in stopping the progression of calcification unless it is started earlier.

In summary, oral administration may represent a simple route to achieve therapeutic levels of the physiological, non-toxic metabolite PPi in the blood circulation. Our data indicate that oral PPi has potential to treat two currently incurable diseases, PXE and GACI. Importantly, oral PPi might have broader applicability and be useful in other conditions involving ectopic calcification, such as hypercholesterolemia (Hoeg *et al*, 1994), diabetes (Kreines *et al*, 1985), chronic renal insufficiency (Moe & Chen, 2004), β-thalassemia (Aessopos *et al*, 1992), and heterotopic ossification of traumatized muscle (Jackson *et al*, 2009). The risk of orally administrated PPi to patients is probably negligible as the WHO considers PPi a non-toxic, physiological, metabolite with a high maximal tolerable daily intake value (MTDI) (http://www.inchem.org/documents/jecfa/jeceval/jec_2259.htm). Moreover, it is extensively used as a food additive, and the Code for Federal Regulation by the FDA states: "This substance is generally recognized as safe when used in accordance with good manufacturing practice" (http://www.accessdata.fda.gov/scripts/cdrh/cfdocs/cfcfr/CFRSearch.cfm?fr = 582.6787). It is also worth noting that PPi would represent a low-cost treatment of connective tissue calcification disorders.

## Materials and Methods

### Human study approval

The human uptake studies were approved by the National Review Board of the Ministry of Health, Hungary (ETT TUKEB). The actual permit based on the above approval has been issued by National Public Health and Medical Officer Service (ÁNTSZ, authorization number: IF-15816-4/2016). Informed consent was obtained from each volunteer prior to the study and experiments conformed to the principles of Declaration of Helsinki what is indicated in the above document. All patient samples were handled in anonymized form also approved by the above document.

### Animals and animal studies

The RCNS, Hungarian Academy of Sciences Institutional Animal Care and Use Committees, approved the animal studies and were conducted according to national guidelines.

C57BL/6J mice designated as wild type were derived from mice purchased from The Jackson Laboratories. *Abcc6$^{-/-}$* mice were generated on 129/Ola background and backcrossed into a C57BL/6J > 10 times. Ttw (*Enpp1$^{-/-}$*) mice (Okawa *et al*, 1998) were bred heterozygous due to the severe phenotype seen in the null animals. Both male and female, age-matched *Abcc6$^{-/-}$*, *Enpp1$^{-/-}$*, and wild-type mice were used. For uptake studies, 15-week-old C57/Bl6 animals were used. In cryo-injury calcification experiments, 12-week-old *Abcc6$^{-/-}$* animals were studied. Calcification of the vibrissae of *Abcc6$^{-/-}$* mice was determined in

22-week-old animals. Calcification of the vibrissae of $Enpp1^{-/-}$ mice was determined in 30-day-old animals. Calcification of the hind limb and kidney arteries of $Enpp1^{-/-}$ mice was quantified in 55-day-old animals.

All animals were housed in approved animal facilities at the Research Centre for Natural Sciences, Hungarian Academy of Sciences. Mice were kept under routine laboratory conditions with 12-hour light–dark cycle with *ad libitum* access to water and chow. Cryo-injury was performed as described previously (Brampton *et al*, 2014).

### Oral uptake of tetrasodium pyrophosphate in humans

Nine or ten healthy volunteers (age 24–69, fasting) ingested a tetrasodium pyrophosphate solution containing 40 mg/kg or 67 mg/kg or 98 mg/kg (43, 72, 110 mM, pH 8.0, respectively). The ingested amounts of tetrasodium pyrophosphate correspond to 13–33% of the maximal tolerable daily intake published by the World Health Organization, WHO (http://www.inchem.org/docu ments/jecfa/jeceval/jec_2259.htm). The duration of ingestion was less than one minute. Blood samples were collected before ingestion (0 min) and 30, 60, 120, 240, 360, and 480 min after from the vena cubiti.

### Pyrophosphate and plasma PPi assay

Sodium pyrophosphate tetrabasic decahydrate (BioXtra quality) was purchased from Sigma and used in animal experiments. For human absorption studies, tetrasodium pyrophosphate anhydrous, Code 118 was purchased from ICL Food Specialist (St. Louis, Missouri, USA). Determination of PPi concentration in plasma was performed as described (Jansen *et al*, 2014). Aliquots from the drinking water during the animal studies were checked for the PPi concentration and found to be stable for at least 4 days. PPi-containing drinking water was changed every second day.

### Ca-measurement

Hearts of $Abcc6^{-/-}$ mice and the tissue blocks harboring the vibrissae of $Enpp1^{-/-}$ mice were digested in 0.15 N HCl for 48 h, and the calcium content was determined by complexometry using the Stanbio Calcium LiquiColor kit (Boerme, TX, USA) following the manufacturer's instructions.

Calcification of the vibrissae of $Abcc6^{-/-}$ mice was quantified by histochemistry as described (Klement *et al*, 2005). Tissue blocks with the hair capsules (vibrissae) were removed, paraffin-embedded, sectioned, and stained with Alizarin Red to visualize calcium deposits. The extent of calcification was quantified by morphometry utilizing image analysis software FIJI (Schindelin *et al*, 2012). The extent of calcification around the vibrissae was quantified by two investigators in a blinded fashion.

Determination of calcification of arteries in the hind limb of $Enpp1^{-/-}$ mice was performed by Alizarin Red staining as described in Kauffenstein *et al* (2014). Individual images of the arteries were combined using Hugin-Panorama photo stitcher (Free Software Foundation, Inc., Boston, MA USA). The resulting images were then processed using ImageMagick (https://www.imagemagick.org).

Kidney tissue sections, 4 μm, were stained by the Yasue procedure as described (Letavernier *et al*, 2016). Sections were

perpendicular to interlobar arteries and 500 μm away from renal hilum. A morphometric analysis was performed (7–13 fields) by using FIJI software (Schindelin *et al*, 2012). Results are expressed as the ratio of calcified area indexed to the whole kidney tissue area.

### Statistical analysis

Data were analyzed by two-tailed Mann–Whitney nonparametric test. Values are expressed as mean ± standard error of the mean (SEM). A $P < 0.05$ was considered statistically significant, and the actual $P$-values are indicated in the corresponding figures. Animal numbers used for individual datasets varied and are shown in the figures.

Expanded View for this article is available online.

### Acknowledgements

The authors are thankful to Piet Borst for his stimulating skepticism at the beginning, for the valuable advice during the study, and for his critical reading of the manuscript. The fruitful discussions with Drs. Balázs Sarkadi, Gergely Szakács, Krisztina Fülöp, and Sharon Terry are also highly appreciated. The technical help of Györgyi Demeter, Emese Törő, and Zsuzsanna Kaminszky is acknowledged. The work was supported by Hungarian grants OTKA 104227, 114336, and VKSz14-1-2015-0155 to A.V., by PXE International to A.V. and K.vd W., and by NIH HL108249 and GM103341 as well as from the Ingeborg v.F. McKee Fund of the Hawaii Community Foundation (15ADVC-74403) to OLS.

### Author contributions

DD, FS, EK, VP, NT, KM, and ET performed experiments and primary data analysis; TRM analyzed data; EL, OLS, and TA designed experiments and analyzed data; KW designed experiments and involved in evaluating data and writing the paper; AV developed the concept, designed experiments, evaluated data, and wrote the article.

---

    

---

### The paper explained

**Problem**

Pyrophosphate (PPi), a natural metabolite, is known to inhibit the pathological calcification of soft tissues including smooth muscle cells of the arteries and several calcification disorders, caused by insufficient levels of PPi. However, it was always assumed that PPi is therapeutically inefficacious when orally taken because its bioavailability is negligible.

**Results**

Orally given PPi appears in the circulation and substantially increases plasma PPi concentrations to levels that are expected to inhibit the soft tissue mineralization seen in several hereditary ectopic calcification disorders. We further show that calcification is strongly attenuated in mouse models of two inherited calcification disorders, pseudoxanthoma elasticum and generalized arterial calcification of infancy, when PPi was provided in the drinking water.

**Impact**

Our results suggest that oral PPi has potential as an effective, simple, and low-cost treatment for patients with conditions involving connective tissue and vascular calcification. The risk of oral administration PPi to patients is probably negligible as it is generally recognized to be safe by FDA.

## Conflict of interest

D.D., F.Sz., K.vdW., and A.V. filed a patent "Oral pyrophosphate for use in reducing tissue calcification" to the Netherland Patent Office (P32885NL00/RKI).

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
