## [Review Process File · EMBO Molecular Medicine]

Oral Administration of Pyrophosphate Inhibits Connective Tissue Calcification

Dora Dedinszki, Flora Szeri, Eszter Kozak, Viola Pomozi, Natalia Tokesi, Tamas Robert Mezei, Kinga Merczel, Emmanuel Letavernier, Ellie Tang, Olivier Le Saux, Tamas Aranyi, Koen van de Wetering, Andras Varadi

Corresponding author: Andras Varadi, Hungarian Academy of Sciences, RCNS

Review timeline:

Submission date:	03 January 2017
Editorial Decision:	08 February 2017
Revision received:	25 April 2017
Editorial Decision:	02 June 2017
Revision received:	09 June 2017
Accepted:	21 June 2017

Transaction Report:

Editor: Roberto Buccione

1st Editorial Decision

08 February 2017

Thank you for the submission of your manuscript to EMBO Molecular Medicine. We are sorry that it has taken longer than we would have liked to get back to you on your manuscript.

You will see that while reviewer 1 is quite positive, reviewer 3 is generally unsupportive of your work. Reviewer 2 is on a middle ground although s/he does raise fundamental concerns that partially overlap with those of reviewer 3. Reviewer 3 suggests that the manuscript's real and only novelty is to propose that oral administration of PPI may be effective as a therapy. On the other hand, this is the very reason why reviewer 1 is so positive. Reviewer 3 (and in part reviewer 2) does not feel that you made a strong enough case in this respect. For instance s/he notes that there is insufficient pharmacokinetic data, both in human and in mice, Reviewer 2 instead notes that oral administration of PPI had only limited effects on the *Ennp1*^{-/-} mouse, which indeed is the appropriate model here.

After our reviewer cross-commenting exercise and in depth internal discussion we have agreed that although conceptual novelty is limited and the overall message is a simple one, we find it of interest. However, given the clear-cut simple message you wish to convey, we all agree that it should be rock solid, which it is not at this stage.

In brief, while publication of the paper cannot be considered at this stage, we would be willing to consider a substantially revised submission, with the understanding that the Reviewers' concerns must be addressed. I thank you for providing an outline of your point-by-point rebuttal and based on it and our discussion, we confirm that we would consider essential for you to perform the following further experimentation, in addition to addressing all the non experimental concerns: 1) Improved

analysis of the effects of oral oral PPI treatment on aorta calcification in *Enpp1*^{-/-} mice, and observation of other soft tissues, 2) detailed pharmacokinetic analysis of orally taken PPI in healthy human volunteers (including more details on the general status of the volunteers) and mice, including time to return to original levels after treatment, 3) analysis of the degradation of PPI in drinking water. We will not be asking you to provide information on the long-term effect of PPI treatment on various plasma parameters, unless you have data at hand.

We remind you that it is EMBO Molecular Medicine policy to allow a single round of revision only and that, therefore, acceptance or rejection of the manuscript will depend on the completeness of your responses included in the next, final version of the manuscript.

As you know, EMBO Molecular Medicine has a "scooping protection" policy, whereby similar findings that are published by others during review or revision are not a criterion for rejection. However, I do ask you to get in touch with us after three months if you have not completed your revision, to update us on the status. Please also contact us as soon as possible if similar work is published elsewhere.

Please note that EMBO Molecular Medicine now requires a complete author checklist (<http://embomolmed.embopress.org/authorguide#editorial3>) to be submitted with all revised manuscripts. Provision of the author checklist is mandatory at revision stage; The checklist is designed to enhance and standardize reporting of key information in research papers and to support reanalysis and repetition of experiments by the community. The list covers key information for figure panels and captions and focuses on statistics, the reporting of reagents, animal models and human subject-derived data, as well as guidance to optimise data accessibility.

We now mandate that all corresponding authors list an ORCID digital identifier. You may do so though our web platform upon submission and the procedure takes < 90 seconds to complete. We also encourage co-authors to supply an ORCID identifier, which will be linked to their name for unambiguous name identification.

Please carefully adhere to our guidelines for authors (<http://embomolmed.embopress.org/authorguide>) to accelerate manuscript processing in case of acceptance.

I look forward to seeing a revised form of your manuscript as soon as possible.

***** Reviewer's comments *****

Referee #1 (Remarks):

The Ms presents an exciting discovery which has considerable potential for treatment of patients with PXE and possibly other calcification disorders associated with low plasma PPI levels. It is a preliminary report demonstrating significant absorption of PPI, in contrast to previous beliefs, with demonstrable attenuation of extra medullary calcification in two animal models of inheritable nature. This is truly novel, remarkable and provides the "first light" regarding mechanism of calcification in PXE as well as having therapeutic potential.

The experiments are well designed. Future investigations will surely further elaborate the pharmacology, therapeutic potential and testing of the interpretation that PPI only attenuates development of calcification and may not affect removal of calcification.

Suggestions:

1. Comment should be made as to what the transport ligand for ABCC6 is...or may be...
2. FIGURE 3 F is not contributory...at least given the scant description of what the "phenotype" is in the mice. Could probably be removed.
3. Editorial scrutiny to correct misuse of tenses, commas and some adverbs!

I consider this as a highly significant study warranting acceptance with minor revision. The Ms will have an important impact....particularly for the many patients with PXE around the world.

Referee #2 (Remarks):

This paper is interesting and novel as it shows that oral administration of PPI can increase the circulating PPI levels both in mouse and humans. There is a potential that by oral PPI treatments, one can prevent ectopic calcification in humans caused by ABCC6 deficiency and perhaps other calcification disorders. However, the mouse model does not fully recapitulate the calcification phenotype of PXE as in mouse the most prominent site of calcification is vibrissae, which is not present in humans. The authors correctly selected another model, the *Enpp1*^{-/-} mice for their study. However, oral administration has only a relative minor effect on the progression of ectopic calcification in this model. I have included specific comments below.

1. The *Enpp1*^{-/-} mouse is a more suitable model for this study as the *Abcc6*^{-/-} mice are not the best model for human ectopic calcification caused by low PPI levels. One weakness of the study lies in the fact that the effects of PPI treatments on the phenotype of *Enpp1*^{-/-} mice is relatively mild. The authors did not show how the initiation and progression of calcification in other more relevant soft tissues (e.g. blood vessel and joints) in *Enpp1*^{-/-} mice are affected by the treatment. These analyses can be performed by histology and some morphometric analyses. It would strengthen the paper significantly if the authors can show that calcification at these sites can be significantly reduced by oral PPI treatment.
2. In Figure 1, it will be important to show the time needed to get the plasma PPI level back to the basal level after the oral administration.
3. Extracellular PPI level is maintained by enzymes like ENPP1 as well as transporter ANK. Surprisingly, ANK and its mode of action were not mentioned in the paper.
4. Page 3: "Inactivating mutations in the genes encoding enzymes involved in PPI homeostasis result in rare hereditary calcification disorders." Please provide references to diseases in addition to GACI and PXE.
5. Gamma glutamyl carboxylase (GGCX) gene dosage has been shown to affect the calcification phenotype of *Abcc6*^{-/-} mice. The current findings need to be discussed in light of this previous finding.
6. In Figure 2, histological analyses with von Kossa staining should be performed to show heart calcification. What is the chemical nature of the deposited mineral? Is it hydroxyapatite or amorphous mineral?
7. Please include scale bars in the histological and tissue images.

Referee #3 (Comments on Novelty/Model System):

The only new information in this study is the oral administration of PPI. The potential beneficial effect of PPI on prevention of vascular calcification is already known.

Referee #3 (Remarks):

In this study, Dedinszki et al show that oral pyrophosphate is absorbed and inhibits tissue calcification. Administering PPI in drinking water in two experimental animal models, the authors observe reduction in calcifications both in *Abcc6*^{-/-} and *Enpp1*^{-/-} mice. In addition, they also note uptake of PPI in humans after oral administration. This is indeed the major message of this paper, since various authors have already shown that the administration of exogenous PPI decreases vascular calcification in experimental models. In addition a number of key issues and concerns remain

1. Title should be more specific. For example, include ... "in mice models". In its present form it could be misleading (they only show effects in mice).
2. μM and mM units should be replaced by $\mu\text{mol/L}$ and mmol/L respectively, both in text and

figures.

3. Please provide more information about the conditions of human experiments: fasting?
4. Since the novelty of this study resides in the via of PPI administration, a more solid pharmacokinetic study should be performed (time course and doses response, both in human and mice).
5. Please provide more information about the conditions of oral administration in mice. For example: in experiment shown in Figure 2C 1), how often was the fresh PPI solution prepared, daily? 2) Does the PPI in water degrade over time?
6. Since PPI could be degraded in the water (figure 2), authors should be analyzed the plasmatic parameters involved in Ca/Pi homeostasis including, mainly, calcium and phosphate levels, and also VitD or PTH levels. Ten mmol / L PPI could be 20 mmol / L Pi and the mice drank over 20 weeks and therefore The reduction in calcium content in tissue could be also explained by alteration (reduction) in calcium homeostasis by long exposition to 10 mmol/L PPI or Pi.
7. Please include more information about calcium determination in Methods section.

1st Revision - authors' response

25 April 2017

Referee #1

1. Comment should be made as to what the transport ligand for ABCC6 is...or may be...

Now we added into the text in the Introduction:

“The liver is the most important source of circulatory PPI, via a pathway depending on ABCC6-mediated ATP release (Jansen et al, 2013; Jansen et al, 2014), **though the exact molecular mechanism of ATP-relapse and the substrate of ABCC6 is not know.**”

2. FIGURE 3 F is not contributory...at least given the scant description of what the "phenotype" is in the mice. Could probably be removed.

We have removed Figure 3F and replaced it with more relevant figure showing the results of calcification of the renal and hind limbs arteries (**New Figure 3**). The figure legend is modified accordingly.

3. Editorial scrutiny to correct misuse of tenses, commas and some adverbs!

We have performed extensive text-editing to avoid the mentioned errors.

Referee #2

1. The Enpp1-/- mouse is a more suitable model for this study as the Abcc6-/- mice are not the best model for human ectopic calcification caused by low PPI levels. One weakness of the study lies in the fact that the effects of PPI treatments on the phenotype of Enpp1-/- mice is relatively mild. The authors did not show how the initiation and progression of calcification in other more relevant soft tissues (e.g. blood vessel and joints) in Enpp1-/- mice are affected by the treatment. These analyses can be performed by histology and some morphometric analyses. It would strengthen the paper significantly if the authors can show that calcification at these sites can be significantly reduced by oral PPI treatment.

We have executed experiments focusing on the calcification of the blood vessels, namely the arteries of the hind limbs (iliac and saphenous arteries...). The results of these experiments are documented in the Results section: “**We detected a robust effect in the extent of calcification inhibition in the hind limb arteries and in the renal arteries. When PPI in as low as 0.3 mM concentration was provided during pregnancy, calcification was reduced to 12% of the levels found in the control group (hind limb arteries) and to 25% (renal arteries), i.e. resulted in 75-88% inhibition (Figure 3, Panels D, E, F, G, H and I).**”

The figure legend is modified accordingly.

2. In Figure 1, it will be important to show the time needed to get the plasma PPI level back to the basal level after the oral administration.

We have completed experiments on oral PPI uptake in human and found as indicated in the Results of the revised manuscript : “**The time needed to get the plasma PPI back to the baseline level**

was 240 minutes at dose 67 mg/kg and 360 minutes at dose 98 mg/kg (Figure 1A). These data indicate a dose- and time-dependent elevation of plasma PPI concentration."

3. *Extracellular PPI level is maintained by enzymes like ENPP1 as well as transporter ANK. Surprisingly, ANK and its mode of action were not mentioned in the paper.*

We accept the referee's criticism and have included a sentence in the revised manuscript: **"Other gene products are also involved in soft tissue calcification affecting PPI homeostasis: like ANK mediating the intracellular to extracellular channeling of PPI (Ho et al, 2000)", though it does not play a role in maintaining plasma PPI.**

(Introduction)

4. *Page 3: "Inactivating mutations in the genes encoding enzymes involved in PPI homeostasis result in rare hereditary calcification disorders." Please provide references to diseases in addition to GACI and PXE.*

We have incorporated the following sentence in the Introduction of the revised manuscript: **"Inactivating mutations in the genes encoding enzymes involved in PPI homeostasis result in rare calcification disorders which include: pseudoxanthoma elasticum (PXE, OMIM 264800), Generalized Arterial Calcification of Infancy (GACI, OMIM 208000), arterial calcification due to CD73 deficiency (ACDC, OMIM 211800), Hutchinson-Gilford Progeria Syndrome (HGPS, OMIM 176670).**

5. *Gamma glutamyl carboxylase (GGCX) gene dosage has been shown to affect the calcification phenotype of Abcc6^{-/-} mice. The current findings need to be discussed in light of this previous finding.*

The referee's point is valid as Vitamin K-dependent glutamate carboxylation and serine phosphorylation convert the non-modified MGP to a protein with calcification inhibitor properties (Schurgers et al, 2007) and *PXE-like calcification disorder with multiple coagulation factor deficiency* is caused by mutation in the gene encoding gamma-glutamyl carboxylase, an enzyme responsible for carboxylation of MGP (Vanakker et al, 2007). However, since the Vitamin-K dependent pathway is not directly related to PPI-action, we decided not to discuss it in the revised manuscript.

6. *In Figure 2, histological analyses with von Kossa staining should be performed to show heart calcification. What is the chemical nature of the deposited mineral? Is it hydroxyapatite or amorphous mineral?*

We have performed histological analysis on calcifying hearts and the images are shown on Figure 2A.

The chemical nature of the deposited mineral is given in the revised manuscript: **"The lesions showed hydroxyapatite crystal nature as determined by transmission electron microscopy (Aherrahrou, 2003)."**

7. *Please include scale bars in the histological and tissue images.*

We have put scale bars on each histological and tissue images (Figures 2 and 3).

Referee #3

1. *Title should be more specific. For example, include ... "in mice models". In its present form it could be misleading (they only show effects in mice).*

Oral Pyrophosphate is Absorbed and Inhibits Connective Tissue Calcification in mice models (90 ch) Unfortunately, this exceeds the character limit of the title. Therefore we can not make this addition (unless the editor agrees with it).

2. *mM and mM units should be replaced by mmol/L and mmol/L respectively, both in text and figures*

It is not clear for us what is the Journal's style

3. *Please provide more information about the conditions of human experiments: fasting?*

The human uptake experiments were performed in fasting conditions, this is indicated now in the Results : **“First we tested whether orally consumed PPI is absorbed in human. Healthy human volunteers (fasting) ingested...”** and this is also stated in the Methods section.

4. *Since the novelty of this study resides in the via of PPI administration, a more solid pharmacokinetic study should be performed (time course and doses response, both in human and mice).*

We have preformed the experiments required by the Referee and the results of those are shown on Figure 1. In the Results section we included the major finding as : **“These data indicate a dose- and time-dependent elevation of plasma PPI concentration. From the data presented in figure 1A we calculated the following pharmacokinetic parameters: $t_{max}=36.7\pm 13.2$ min, $C_{max}=3.9\pm 1.6$ μ M, $t_{1/2}=44.7\pm 16.7$ min (single exponential decay) when 98 mg Na4PPI per kg body weight was given.”** (in human). In mouse: **“... we followed the uptake of PPI (50 mM, 200 μ L) delivered directly to the stomach over time. PPI was rapidly absorbed from the stomach (Figure 1D) and, as expected, its plasma concentrations depended on the dose given (Figure 1E).”**

5. *Please provide more information about the conditions of oral administration in mice. For example: in experiment shown in Figure 2C 1), how often was the fresh PPI solution prepared, daily? 2) Does the PPI in water degrade over time?*

This information is now detailed in the Methods section of the revised manuscript as **“Aliquots from the drinking water during the animal studies were checked for the PPI concentration and found to be stable for at least 4 days. PPI containing drinking water was changed every second day.”**

6. *Since PPI could be degraded in the water (figure 2), authors should be analyzed the plasmatic parameters involved in Ca/Pi homeostasis including, mainly, calcium and phosphate levels ,and also VitD or PTH levels. Ten mmol / L PPI could be 20 mmol / L Pi and the mice drank over 20 weeks and therefore The reduction in calcium content in tissue could be also explained by alteration (reduction) in calcium homeostasis by long exposition to 10 mmol/L PPI or Pi.*

We have agreed with the Editor in the consultation letter that that Ca, Pi and VitD or PTH levels will not be subjected to analyses in the present study.

7. *Please include more information about calcium determination in Methods section.*

Detailed information is given now in the Methods section on both types of calcium measurement (complexometry and histochemistry combined with morphometry) in the Methods section.

“Ca-measurement

Hearts of *Abcc6*^{-/-} mice and the tissue blocks harboring the vibrissae of *Enpp1*^{-/-} mice were digested in 0.15 N HCl for 48 hours and the calcium content was determined by complexometry using the Stanbio Calcium LiquiColor kit (Boerne, TX, USA) following the manufacturer’s instructions.

Calcification of the vibrissae of *Abcc6*^{-/-} mice was quantified by histochemistry as described (Klement et al, 2005). Tissue blocks with the hair capsules (vibrissae) were removed, paraffin-embedded, sectioned and stained with Alizarin Red to visualize calcium deposits. The extent of calcification was quantified by morphometry utilizing image analysis software FIJI (Schindelin et al, 2012). The extent of calcification around the vibrissae was quantified by two investigators in a blinded fashion.

Determination of calcification of arteries in the hind limb of *Enpp1*^{-/-} mice was performed by Alizarin Red staining as described in Kauffenstein et al, 2014. Individual images of the arteries were combined using Hugin-Panorama photo stitcher (Free Software Foundation, Inc., Boston, MA USA). The resulting images were then processed using ImageMagick (<https://www.imagemagick.org>).

Kidney tissue sections, 4 μ m, were stained by the Yasue procedure as described (Letavernier et al, 2016). Sections were perpendicular to interlobar arteries and 500 μ m away from renal hilum. A morphometric analysis was performed (7- 13 fields) by using FIJI software (Schindelin et al, 2012). Results are expressed as the ratio of calcified area indexed to the whole kidney tissue area.”

Thank you for the submission of your revised manuscript to EMBO Molecular Medicine.

I asked reviewers 2 and 3 to re-evaluate your revised manuscript. We have now received the enclosed reports from reviewer 2. Unfortunately I failed to obtain a re-evaluation from reviewer 3. As a further delay cannot be justified, I am proceeding with the available evaluation. As you will see reviewer 2 is now globally supportive. As for reviewer 3, we have now considered your rebuttal at the editorial level, and found your actions and replies to be satisfactory and to address his/her concerns.

I am therefore pleased to inform you that we will be able to accept your manuscript pending the following final amendments:

- 1) Please add "P=" to all P values in the figures to increase comprehension.
- 2) In response to your query on the title, I would not worry about specifying further since after all you do verify absorption in humans. Rather, I would like to propose an alternative title (see attached modified manuscript).
- 3) In response to your query on figure 1 panels, I would leave as they are.
- 4) In response to your query on how to cite the AJP paper, this is no longer a concern as it is now published.
- 5) Please remove the bullet points from the reference list.
- 6) The manuscript must include a statement in the Materials and Methods identifying the institutional and/or licensing committee approving the experiments, including any relevant details (like how many animals were used, of which gender, at what age, which strains, if genetically modified, on which background, housing details, etc). We encourage authors to follow the ARRIVE guidelines for reporting studies involving animals. Please see the EQUATOR website for details: <http://www.equator-network.org/reporting-guidelines/improving-bioscience-research-reporting-the-arrive-guidelines-for-reporting-animal-research/>. Please make sure that ALL the above details are reported both in the manuscript and the checklist.
- 7) For experiments involving human subjects the authors must identify the committee approving the experiments and include a statement that informed consent was obtained from all subjects and that the experiments conformed to the principles set out in the WMA Declaration of Helsinki [<http://www.wma.net/en/30publications/10policies/b3/>] and the NIH Belmont Report [<http://ohsr.od.nih.gov/guidelines/belmont.html>]. Any restrictions on the availability or on the use of human data or samples should be clearly specified in the manuscript. Any restrictions that may detract from the overall impact of a study or undermine its reproducibility will be taken into account in the editorial decision. Please make sure that ALL the above details are reported both in the manuscript and the checklist. Furthermore, please make sure that the details reported in the checklist (e.g. authorization number) are also reported in the manuscript.
- 8) We encourage the publication of source data, with the aim of making primary data more accessible and transparent to the reader. Would you be willing to provide a PDF file per figure that contains the original, uncropped and unprocessed scans of all or at least the key gels used in the manuscript and/or source data sets for relevant graphs? The files should be labeled with the appropriate figure/panel number, and in the case of gels, should have molecular weight markers; further annotation may be useful but is not essential. The files will be published online with the article as supplementary "Source Data" files. If you have any questions regarding this just contact me.
- 9) I have gone through your text and made some suggested changes (see attached) in the Title, Abstract and The Paper Explained sections of your manuscript to improve readability and impact. I would appreciate it if you could work from this version when preparing your revision. If you have any problems opening the file or tracking the changes, please let me know. I would also recommend

a final read by a native English speaker to weed out a few remaining grammar/spelling issues that I do not have the time to deal with.

Please submit your revised manuscript within two weeks. I look forward to seeing a revised form of your manuscript as soon as possible.

***** Reviewer's comments *****

Referee #2 (Comments on Novelty/Model System):

All critical aspects I have raised were appropriately addressed.

Corresponding Author Name: András Váradi

Manuscript Number: EMM-2017-07532